# PEP: Parameter Ensembling by Perturbation

**Alireza Mehrtash**[1,2], **Purang Abolmaesumi**[1], **Polina Golland**[3],
**Tina Kapur**[2], **Demian Wassermann**[4], **William M. Wells III**[2,3]

[1]ECE Department, University of British Columbia (UBC), Vancouver, BC
[2]Department of Radiology, BWH, Harvard Medical School, Boston, MA
[3]CSAIL, MIT, Boston, MA    [4]INRIA Saclay, Palaiseau, France

`{mehrtash,sw}@bwh.harvard.edu`

## Abstract

Ensembling is now recognized as an effective approach for increasing the predictive performance and calibration of deep networks. We introduce a new approach, Parameter Ensembling by Perturbation (PEP), that constructs an ensemble of parameter values as random perturbations of the optimal parameter set from training by a Gaussian with a single variance parameter. The variance is chosen to maximize the log-likelihood of the ensemble average ($\mathbb{L}$) on the validation data set. Empirically, and perhaps surprisingly, $\mathbb{L}$ has a well-defined maximum as the variance grows from zero (which corresponds to the baseline model). Conveniently, calibration level of predictions also tends to grow favorably until the peak of $\mathbb{L}$ is reached. In most experiments, PEP provides a small improvement in performance, and, in some cases, a substantial improvement in empirical calibration. We show that this "PEP effect" (the gain in log-likelihood) is related to the mean curvature of the likelihood function and the empirical Fisher information. Experiments on ImageNet pre-trained networks including ResNet, DenseNet, and Inception showed improved calibration and likelihood. We further observed a mild improvement in classification accuracy on these networks. Experiments on classification benchmarks such as MNIST and CIFAR-10 showed improved calibration and likelihood, as well as the relationship between the PEP effect and overfitting; this demonstrates that PEP can be used to probe the level of overfitting that occurred during training. In general, no special training procedure or network architecture is needed, and in the case of pre-trained networks, no additional training is needed.

## 1   Introduction

Deep neural networks have achieved remarkable success on many classification and regression tasks [28]. In the usual usage, the parameters of a conditional probability model are optimized by maximum likelihood on large amounts of training data [10]. Subsequently the model, in combination with the optimal parameters, is used for inference. Unfortunately, this approach ignores uncertainty in the value of the estimated parameters; as a consequence over-fitting may occur and the results of inference may be overly confident. In some domains, for example medical applications, or automated driving, overconfidence can be dangerous [1].

Probabilistic predictions can be characterized by their level of *calibration*, an empirical measure of consistency with outcomes, and work by Guo et al. shows that modern neural networks (NN) are often poorly calibrated, and that a simple one-parameter *temperature scaling* method can improve their calibration level [12]. Explicitly Bayesian approaches such as *Monte Carlo Dropout* (MCD) [8] have been developed that can improve likelihoods or calibration. MCD approximates a Gaussian process at inference time by running the model several times with active dropout layers. Similar to the

MCD method [8], Teye et al. [45] showed that training NNs with batch normalization (BN) [18] can be used to approximate inference with Bayesian NNs. Directly related to the problem of uncertainty estimation, several works have studied out-of-distribution detection. Hendrycks and Gimpel [14] used softmax prediction probability baseline to effectively predict misclassification and out-of-distribution in test examples. Liang et al. [31] used temperature scaling and input perturbations to enhance the baseline method of Hendrycks and Gimpel [14]. In a recent work, Rohekar et al. [39] proposed a method for confounding training in deep NNs by sharing neural connectivity between generative and discriminative components. They showed that using their BRAINet architecture, which is a hierarchy of deep neural connections, can improve uncertainty estimation. Hendrycks et al. [15] showed that using pre-training can improve uncertainty estimation. Thulasidasan et al. [46] showed that mixed up training can improve calibration and predictive uncertainty of models. Corbière et al. [5] proposed *True Class Probability* as an alternative for classic Maximum Class Probability. They showed that learning the proposed criterion can improve model confidence and failure prediction. Raghu et al. [37] proposed a method for direct uncertainty prediction that can be used for medical second opinions. They showed that deep NNs can be trained to predict uncertainty scores of data instances that have high human reader disagreement.

Ensemble methods [6] are regarded as a straightforward way to increase the performance of base networks and have been used by the top performers in imaging challenges such as ILSVRC [44]. The approach typically prepares an ensemble of parameter values that are used at inference-time to make multiple predictions, using the same base network. Different methods for ensembling have been proposed for improving model performance, such as M-heads [30] and Snapshot Ensembles [16]. Following the success of ensembling methods in improving baseline performance, Lakshminarayanan et al. proposed *Deep Ensembles* in which model averaging is used to estimate predictive uncertainty [26]. By training collections of models with random initialization of parameters and adversarial training, they provided a simple approach to assess uncertainty.

Deep Ensembles and MCD have both been successfully used in several applications for uncertainty estimation and calibration improvement. However, Deep Ensembles requires retraining a model from scratch for several rounds, which is computationally expensive for large datasets and complex models. Moreover, Deep Ensembles cannot be used to calibrate pre-trained networks for which the training data is not available. MCD requires the network architecture to have dropout layers, hence there is a need for network modification if the original architecture does not have dropout layers. In many modern networks, BN removes the need for dropout [18]. It is also challenging or not feasible in some cases to use MCD on out-of-the-box pre-trained networks.

Gaussians are an attractive choice of distribution for going beyond point estimates of network parameters – they are easily sampled to approximate the marginalization that is needed for predictions, and the Laplace approximation can be used to characterize the covariance by using the Hessian of the loss function. Kristiadi et al. [23] support this approach for mitigating the overconfidence of ReLU-based networks. They use a Laplace approximation that is based on the last layer of the network that provides improvements to predictive uncertainty and observe that "a sufficient condition for a calibrated uncertainty on a ReLU network is to be a bit Bayesian." Ritter et al. [38] use a Laplace approach with a layer-wise Kronecker factorization of the covariance that scales only with the square of the size of network layers and obtain improvements similar to dropout. Izmailov et al. [19] describe a stochastic weight averaging Stochastic Weight Averaging (SWA) approach that averages in weight space rather than in model space such as ensembling approaches and approaches that sample distributions on parameters. Averages are calculated over weights observed during training via SGD, leading to wider optima and better generalization in experiments on CIFAR10, CIFAR100 and ImageNet. Building on SWA, Maddox et al. [32] describe Stochastic Weight Averaging-Gaussian (SWAG) that constructs a Gaussian approximation to the posterior on weights. It uses SWA to estimate the first moment on weights combined with a low-rank plus diagonal covariance estimate. They show that SWAG is useful for out of sample detection, calibration and transfer learning.

In this work, we propose Parameter Ensembling by Perturbation (PEP) for deep learning, a simple ensembling approach that uses random perturbations of the optimal parameters from a single training run. PEP is perhaps the simplest possible Laplace approximation - an isotropic Gaussian with one variance parameter, though we set the parameter with simple ML/cross-validation rather than by calculating curvature. Parameter perturbation approaches have been previously used in climate research [33, 2] and they have been used to good effect in variational Bayesian deep learning [21] and to improve adversarial robustness [20].

Unlike MCD which needs dropout at training, PEP can be applied to any pre-trained network without restrictions on the use of dropout layers. Unlike Deep Ensembles, PEP needs only one training run. PEP can provide improved log-likelihood and calibration for classification problems, without the need for specialized or additional training, substantially reducing the computational expense of ensembling. We show empirically that the log-likelihood of the ensemble average ($\mathbb{L}$) on hold-out validation and test data grows initially from that of the baseline model to a well-defined peak as the spread of the parameter ensemble increases. We also show that PEP may be used to probe curvature properties of the likelihood landscape. We conduct experiments on deep and large networks that have been trained on ImageNet (ILSVRC2012) [40] to assess the utility of PEP for improvements on calibration and log-likelihoods. The results show that PEP can be used for probability calibration on pre-trained networks such as DenseNet [17], Inception [44], ResNet [13], and VGG [43]. Improvements in log-likelihood range from small to significant but they are almost always observed in our experiments. To compare PEP with MCD and Deep Ensembles, we ran experiments on classification benchmarks such as MNIST and CIFAR-10 which are small enough for us to re-train and add dropout layers. We carried out an experiment with non-Gaussian perturbations We performed further experiments to study the relationship between over-fitting and the "PEP effect," (the gain in log likelihood over the baseline model) where we observe larger PEP effects for models with higher levels of over-fitting, and finally, we showd that PEP can improve out-of-distribution detection.

To the best of our knowledge, this is the first report of using ensembles of perturbed deep nets as an accessible and computationally inexpensive method for calibration and performance improvement. Our method is potentially most useful when the cost of training from scratch is too high in terms of effort or carbon footprint.

## 2 Method

In this section, we describe the PEP model and analyze local properties of the resulting PEP effect (the gain in log-likelihood over the comparison baseline model). In summary PEP is formulated in the Bayes' network (hierarchical model) framework; it constructs ensembles by Gaussian perturbations of the optimal parameters from training. The single variance parameter is chosen to maximize the likelihood of ensemble average predictions on validation data, which, empirically, has a well-defined maximum. PEP can be applied to any pre-trained network; only one standard training run is needed, and no special training or network architecture is needed.

### 2.1 Baseline Model

We begin with a standard discriminative model, e.g., a classifier that predicts a distribution on $y_i$ given an observation $x_i$,

$$p(y_i; x_i, \theta) \ . \tag{1}$$

Training is conventionally accomplished by maximum likelihood,

$$\theta^* \doteq \underset{\theta}{\operatorname{argmax}} \, \mathcal{L}(\theta) \qquad \text{where the log-likelihood is:} \qquad \mathcal{L}(\theta) \doteq \sum_i \ln L_i(\theta) \ , \tag{2}$$

and $L_i(\theta) \doteq p(y_i; x_i, \theta)$ are the individual likelihoods. Subsequent predictions are made with the model using $\theta^*$.

### 2.2 Hierarchical Model

Empirically, different optimal values of $\theta$ are obtained on different data sets; we aim to model this variability with a very simple parametric model – an isotropic normal distribution with mean and scalar variance parameters,

$$p(\theta; \bar{\theta}, \sigma) \doteq N(\theta; \bar{\theta}, \sigma^2 I) \ . \tag{3}$$

The product of Eqs. 1 and 3 specifies a joint distribution on $y_i$ and $\theta$; from this we can obtain model predictions by marginalizing over $\theta$, which leads to

$$p(y_i; x_i, \bar{\theta}, \sigma) = \mathbb{E}_{\theta \sim N(\bar{\theta}, \sigma^2 I)} \left[ p(y_i; x_i, \theta) \right] \ . \tag{4}$$

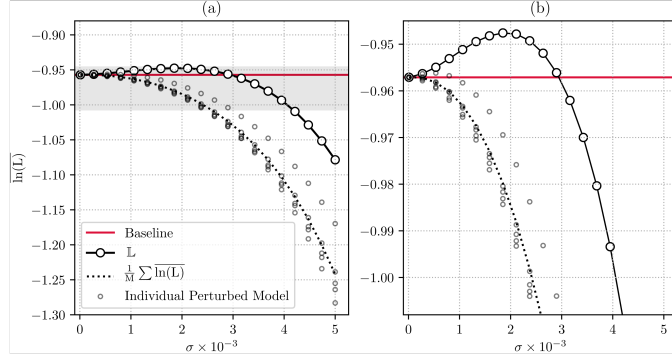

Figure 1: Parameter Ensembling by Perturbation (PEP) on pre-trained InceptionV3 [44]. The rectangle shaded in gray in (a) is shown in greater detail in (b). The average log-likelihood of the ensemble average, $\mathbb{L}(\sigma)$, has a well-defined maximum at $\sigma = 1.85 \times 10^{-3}$. The ensemble also has a noticeable increase in likelihood over the individual ensemble item average log-likelihoods, $\overline{\ln(L)}$ and over their average. In this experiment, an ensemble size of 5 (M = 5) was used for PEP and the experiments were run on 5000 validation images.

We approximate the expectation by a sample average,

$$p(y_i; x_i, \bar{\theta}, \sigma) \approx \frac{1}{m} \sum_j p(y_i; x_i, \theta_j) \qquad \text{where} \qquad \theta_{j=1}^m \underset{\text{IID}}{\leftarrow} N(\bar{\theta}, \sigma^2 I), \tag{5}$$

i.e., the predictions are made by averaging over the predictions of an ensemble. The log-likelihood of the ensemble prediction as a function of $\sigma$ is then

$$\mathbb{L}(\sigma) \doteq \sum_i \ln \frac{1}{m} \sum_j L_i(\theta_j) \qquad \text{where} \qquad \theta_{j=1}^m \underset{\text{IID}}{\leftarrow} N(\bar{\theta}, \sigma^2 I) \tag{6}$$

(dependence on $\bar{\theta}$ is suppressed for clarity). Throughout most of paper we will use $i$ to index data items, $j$ to index ensemble of parameters, and $m$ to indicate the size of the ensemble. We estimate the model parameters as follows. First we optimize $\theta$ with $\sigma$ fixed at zero using a training data set (when $\sigma \to 0$ the $\theta_j \to \bar{\theta}$), then

$$\theta^* = \underset{\bar{\theta}}{\text{argmax}} \sum_i \ln p(y_i; x_i, \bar{\theta}) , \tag{7}$$

which is equivalent to maximum likelihood parameter estimation of the base model. Next we optimize over $\sigma$, (using a validation data set), with $\theta$ fixed at the previous estimate, $\theta^*$,

$$\sigma^* = \underset{\sigma}{\text{argmax}} \sum_i \ln \frac{1}{m} \sum_{\theta_j} p(y_i; x_i, \theta_j) \ \text{where} \ \theta_{j=1}^m \underset{\text{IID}}{\leftarrow} N(\theta^*, \sigma^2 I) . \tag{8}$$

Then at test time the ensemble prediction is

$$p(y_i; x_i, \theta^*, \sigma^*) \approx \frac{1}{m} \sum_{\theta_j} p(y_i; x_i, \theta_j) \ \text{where} \ \theta_{j=1}^m \underset{\text{IID}}{\leftarrow} N(\theta^*, \sigma^{*2} I) . \tag{9}$$

In our experiments, perhaps somewhat surprisingly, $\mathbb{L}(\sigma)$ has a well-defined maximum away from $\sigma = 0$ (which corresponds to the baseline model). As $\sigma$ grows from 0, $\mathbb{L}(\sigma)$ rises to a well-defined peak value, then falls dramatically (Figure 1). Conveniently, the calibration quality tends to grow favorably until the $\mathbb{L}(\sigma)$ peak is reached. It may be that $\mathbb{L}(\sigma)$ initially grows because the classifiers corresponding to the ensemble parameters remain accurate, and the ensemble performs better as the classifiers become more independent [6]. Figure 1 shows $\mathbb{L}(\sigma)$ for experiments with InceptionV3 [44], along with the average log-likelihoods ($\overline{\ln(L)}$) of the individual ensemble members. Note that in the figures, in the current machine learning style, we have used averaged log-likelihoods, while in

this section we use the estimation literature convention that log-likelihoods are summed rather than averaged. We can see that for several members, $\overline{\ln(L)}$ grows somewhat initially, this indicates that the optimal parameter from training is not optimal for the validation data. Interestingly, the ensemble has a more robust increase, which persists over scale substantially longer than for the individual networks. We have observed this $\mathbb{L}(\sigma)$ "increase to peak" phenomenon in many experiments with a wide variety of networks.

## 2.3 Local Analysis

In this section, we analyze the nature of the PEP effect in the neighborhood of $\theta^*$. Returning to the log-likelihood of a PEP ensemble (Eq. 6), and "undoing" the approximation by sample average,

$$\mathbb{L}(\sigma) \approx \sum_i \ln \mathbb{E}_{\theta \sim N(\theta^*, \sigma^2 I)} \left[ L_i(\theta) \right] . \tag{10}$$

Next, we develop a local approximation to the expected value of the log-likelihood. The following formula is derived in the Appendix (Eq 5) using a second-order Taylor expansion about the mean.

For $x \sim N(\mu, \Sigma)$

$$\mathbb{E}_x \left[ f(x) \right] \approx f(\mu) + \frac{1}{2} T_R(Hf(\mu)\Sigma) , \tag{11}$$

where $Hf(x)$ is the Hessian of $f(x)$ and $T_R$ is the trace. In the special case that $\Sigma = \sigma^2 I$,

$$\mathbb{E}_x \left[ f(x) \right] \approx f(\mu) + \frac{\sigma^2}{2} \triangle f(\mu) \tag{12}$$

where $\triangle$ is the Laplacian, or mean curvature. The appendix shows that the third Taylor term vanishes due to Gaussian properties, so that the approximation residual is $O(\sigma^4 \partial^4 f(\mu))$ where $\partial^4$ is a specific fourth derivative operator.

Applying this to the log-likelihood in Eq. 10 yields

$$\mathbb{L}(\sigma) \approx \sum_i \ln \left[ L_i(\theta^*) + \frac{\sigma^2}{2} \triangle L_i(\theta^*) \right] \approx \sum_i \left[ \ln L_i(\theta^*) + \frac{\sigma^2}{2} \frac{\triangle L_i(\theta^*)}{L_i(\theta^*)} \right] \tag{13}$$

(to first order), or

$$\mathbb{L}(\sigma) \approx \mathcal{L}(\theta^*) + B_\sigma(\theta^*) , \tag{14}$$

where $\mathcal{L}(\theta)$ is the log-likelihood of the base model (Eq. 2) and

$$B_\sigma(\theta) \doteq \frac{\sigma^2}{2} \sum_i \frac{\triangle L_i(\theta)}{L_i(\theta)} \tag{15}$$

is the "PEP effect." Note that its value may be dominated by data items that have low likelihood, perhaps because they are difficult cases, or incorrectly labeled. Next we establish a relationship between the PEP effect and the Laplacian of the log-likelihood of the base model. From Appendix (Eq 34) ,

$$\triangle\mathcal{L}(\theta) = \sum_i \left[ \frac{\triangle L_i(\theta)}{L_i(\theta)} - (\nabla \ln L_i(\theta))^2 \right] \tag{16}$$

(here the square in the second term on the right is the dot product of two gradients) Then

$$\triangle\mathcal{L}(\theta) = \frac{2}{\sigma^2} B_\sigma(\theta) - \sum_i (\nabla \ln L_i(\theta))^2 \tag{17}$$

or

$$B_\sigma(\theta) = \frac{\sigma^2}{2} \left[ \triangle\mathcal{L}(\theta) + \sum_i (\nabla \ln L_i(\theta))^2 \right] . \tag{18}$$

The empirical Fisher information (FI) is defined in terms of the outer product of gradients as

$$\widetilde{F}(\theta) \doteq \sum_i \nabla \ln L_i(\theta) \nabla \ln L_i(\theta)^T \tag{19}$$

(see [25]) . So, the second term above in Eq. 18 is the trace of the empirical FI. Then finally the PEP effect can be expressed as

$$B_\sigma(\theta) = \frac{\sigma^2}{2} \left[ \triangle \mathcal{L}(\theta) + T_R(\widetilde{F}(\theta)) \right] \ . \tag{20}$$

The first term of the PEP effect in Eq. 20, the mean curvature of the log-likelihood, can be positive or negative, (we expect it to be negative near the mode), while the second term, the trace of the empirical Fisher information, is non-negative. As the sum of squared gradients, we may expect the second term to grow as $\theta$ moves away from the mode.

The first term may also be seen as a (negative) trace of an empirical FI. If the sum is converted to an average it approximates an expectation that is equal to the negative of the trace of the Hessian form of the FI, while the second term is the trace of a different empirical FI. Empirical FI are said to be most accurate at the mode of the log-likelihood [25]. So, if $\theta^*$ is close to the log-likelihood mode on the new data, we may expect the terms to cancel. If $\theta^*$ is farther from the log-likelihood mode on the new data, they may no longer cancel.

Next, we discuss two cases, in both we examine the log-likelihood of the validation data, $\mathcal{L}(\theta)$, at $\theta^*$, the result of optimization on the training data. In general, $\theta^*$ will not coincide with the mode of the log-likelihood of the validation data. **Case 1:** $\theta^*$ is 'close' to the mode of the validation data, so we expect the mean curvature to be negative. **Case 2**: $\theta^*$ is 'not close' to the mode of the validation data, so the mean curvature may be positive. We conjecture that case 1 characterizes the likelihood landscape on new data when the baseline model is not overfitted, and that case 2 is characteristic of an overfitted model (where, empirically, we observe positive PEP effect).

As these are local characterizations, they are only valid near $\theta^*$. While the analysis may predict PEP effect for small $\sigma$, as it grows, and the $\theta_j$ move farther from the mode, the log-likelihood will inevitably decrease dramatically (and there will be a peak value between the two regimes).

There has been a lot of work recently concerning the curvature properties of the log-likelihood landscape. Gorbani et al. point out that "Hessian of training loss ... is crucial in determining many behaviors of neural networks"; they provide tools to analyze the Hessian spectrum and point out characteristics associated with networks trained with BN [9]. Sagun et al. [41] show that there is a 'bulk' of zero valued eigenvalues of the Hessian that can be used to analyze overparameterization, and in a related paper discuss implications that "shed light on the geometry of high-dimensional and non-convex spaces in modern applications" [42]. Goodfellow et al. [11] report on experiments that characterize the loss landscape by interpolating among parameter values, either from the initial to final values or between different local minima. Some of these demonstrate convexity of the loss function along the line segment, and they suggest that the optimization problems are less difficult than previously thought. Fort et al. [7] analyze Deep Ensembles from the perspective of the loss landscape, discussing multiple modes and associated connectors among them. While the entire Hessian spectrum is of interest, some insights may be gained from the avenues to characterizing the mean curvature that PEP provides.

## 3 Experiments

This section reports performance of PEP, and compares it to temperature scaling [12], MCD [8], and Deep Ensembles [26], as appropriate. The first set of results are on ImageNet pre-trained networks where the only comparison is with temperature scaling (no training of the baselines was carried out so MCD and Deep Ensembles were not evaluated). Then we report performance on smaller networks, MNIST and CIFAR-10, where we compare to MCD and Deep Ensembles as well. We also show that the PEP effect is strongly related to the degree of overfitting of the baseline networks.

**Evaluation metrics:** Model calibration was evaluated with negative log-likelihood (NLL), Brier score [3] and reliability diagrams [34]. NLL and Brier score are proper scoring rules that are commonly used for measuring the quality of classification uncertainty [36, 26, 8, 12]. Reliability diagrams plot expected accuracy as a function of class probability (confidence), and perfect calibration is achieved when confidence (x-axis) matches expected accuracy (y-axis) exactly [34, 12]. Expected Calibration Error (ECE) is used to summarize the results of the reliability diagram. Details of evaluation metrics are given in the Supplementary Material (Appendix B).

Table 1: ImageNet results: For all models except VGG19, PEP achieves statistically significant improvements in calibration compared to baseline (BL) and temperature scaling (TS), in terms of NLL and Brier score. PEP also reduces test errors, while TS does not have any effect on test errors. Although TS and PEP outperform baseline in terms of ECE% for DenseNet121, DenseNet169, ResNet, and VGG16, the improvements in ECE% is not consistent among the methods. $T^*$ and $\sigma^*$ denote optimized temperature for TS and optimized sigma for PEP, respectively. Boldfaced font indicates the best results for each metric of a model and shows that the differences are statistically significant ($p$-value$<0.05$).

| Model | $T^*$ | $\sigma^*$ $\times 10^{-3}$ | Negative log-likelihood | | | Brier score | | | ECE% | | | Top-1 error % | |
|---|---|---|---|---|---|---|---|---|---|---|---|---|---|
| | | | BL | TS | PEP | BL | TS | PEP | BL | TS | PEP | BL | PEP |
| DenseNet121 | 1.10 | 1.94 | 1.030 | 1.018 | **0.997** | 0.357 | 0.356 | **0.349** | 3.47 | **1.52** | 2.03 | 25.73 | **25.13** |
| DenseNet169 | 1.23 | 2.90 | 1.035 | 1.007 | **0.940** | 0.354 | 0.350 | **0.331** | 5.47 | **1.75** | 2.35 | 25.31 | **23.74** |
| IncepttionV3 | 0.91 | 1.94 | 0.994 | 0.975 | **0.950** | 0.328 | 0.328 | **0.317** | **1.80** | 4.19 | 2.46 | 22.96 | **22.26** |
| ResNet50 | 1.19 | 2.60 | 1.084 | 1.057 | **1.023** | 0.365 | 0.362 | **0.350** | 5.08 | **1.97** | 2.94 | 26.09 | **25.18** |
| VGG16 | 1.09 | 1.84 | 1.199 | 1.193 | **1.164** | 0.399 | 0.399 | **0.391** | 2.52 | 2.08 | **1.64** | 29.39 | **28.83** |
| VGG19 | 1.09 | 1.03 | 1.176 | 1.171 | 1.165 | 0.394 | 0.394 | 0.391 | 4.77 | 4.50 | 4.48 | 28.99 | 28.75 |

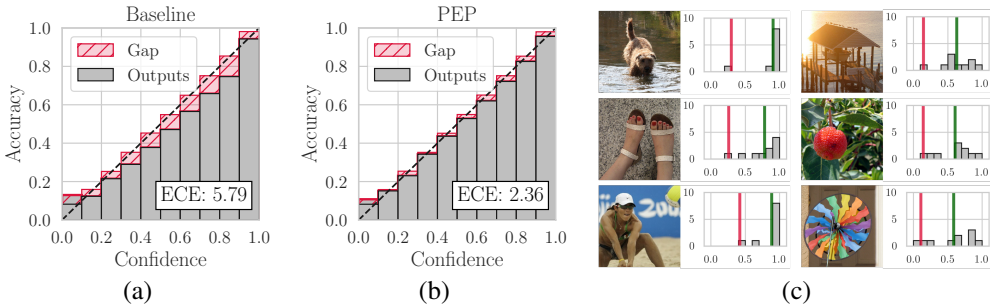

(a)   (b)   (c)

Figure 2: Improving pre-trained DenseNet169 with PEP (M=10). (a) and (b) show the reliability diagrams of the baseline and the PEP. (c) shows examples of misclassifications corrected by PEP. The examples were among those with the largest PEP effect on the correct class probability. (c) Top row: brown bear and lampshade changed into Irish terrier and boathouse; Middle row: band aid and pomegranate changed into sandal and strawberry; Bottom row: bathing cap and wall clock changed into volleyball and pinwheel. The histograms at the right of each image illustrate the probability distribution of ensemble. Vertical red and green lines show the predicted class probabilities of the baseline and the PEP for the correct class label. (For more reliability diagrams see Supplementary Material.)

## 3.1   ImageNet experiments

We evaluated the performance of PEP using large scale networks that were trained on ImageNet (ILSVRC2012) [40] dataset. We used the subset of 50,000 validation images and labels that is included in the development kit of ILSVRC2012. From the 50,000 images, 5,000 images were used as a validation set for optimizing $\sigma$ in PEP, and temperature $T$ in temperature scaling. The remaining 45,000 images were used as the test set. Golden section search [35] was used to find the $\sigma^*$ that maximizes $\mathbb{L}(\sigma)$. The search range for $\sigma$ was $5\times10^{-5}$–$5\times10^{-3}$, ensemble size was 5 ($m = 5$), and number of iterations was 7. On the test set with 45,000 images, PEP was evaluated using $\sigma^*$ and with ensemble size of 10 ($m = 10$). Single crops of the center of images were used for the experiments. Evaluation was performed on six pre-trained networks from the Keras library[4]: DenseNet121, DenseNet169 [17], InceptionV3 [44], ResNet50 [13], VGG16, and VGG19 [43]. For all pre-trained networks, Gaussian perturbations were added to the weights of all convolutional layers. Table 1 summarizes the optimized $T$ and $\sigma$ values, model calibration in terms of NLL, Brier score, and classification errors. For all the pre-trained networks, except VGG19, PEP achieves statistically significant improvements in calibration compared to the baseline and temperature scaling. Note the reduction in top-1 error of DenseNet169 by about 1.5 percentage points, and the reduction in all top-1 errors. Figure 2 shows the reliability diagram for DenseNet169, before and after calibration with PEP with some corrected misclassification examples.

Table 2: MNIST, Fashion MNIST, CIFAR-10, and CIFAR-100 results. The table summarizes experiments described in Section 3.2.

| Experiment | Baseline | PEP | Temp. Scaling | MCD | SWA | Deep Ensembles |
|---|---|---|---|---|---|---|
| | | | NLL | | | |
| MNIST (MLP) | $0.096 \pm 0.01$ | $0.079 \pm 0.01$ | $0.074 \pm 0.01$ | $0.094 \pm 0.00$ | $0.067 \pm 0.00$ | $0.044 \pm 0.00$ |
| MNIST (CNN) | $0.036 \pm 0.00$ | $0.034 \pm 0.00$ | $0.032 \pm 0.00$ | $0.031 \pm 0.00$ | $0.028 \pm 0.00$ | $0.021 \pm 0.00$ |
| Fashion MNIST | $0.360 \pm 0.01$ | $0.275 \pm 0.01$ | $0.271 \pm 0.01$ | $0.218 \pm 0.01$ | $0.277 \pm 0.01$ | $0.198 \pm 0.00$ |
| CIFAR-10 | $1.063 \pm 0.03$ | $0.982 \pm 0.02$ | $0.956 \pm 0.02$ | $0.798 \pm 0.01$ | $0.827 \pm 0.01$ | $0.709 \pm 0.00$ |
| CIFAR-100 | $2.685 \pm 0.03$ | $2.651 \pm 0.03$ | $2.606 \pm 0.03$ | $2.435 \pm 0.03$ | $2.314 \pm 0.02$ | $2.159 \pm 0.01$ |
| | | | Brier | | | |
| MNIST (MLP) | $0.037 \pm 0.00$ | $0.035 \pm 0.00$ | $0.035 \pm 0.00$ | $0.040 \pm 0.00$ | $0.032 \pm 0.00$ | $0.020 \pm 0.00$ |
| MNIST (CNN) | $0.016 \pm 0.00$ | $0.015 \pm 0.00$ | $0.015 \pm 0.00$ | $0.014 \pm 0.00$ | $0.013 \pm 0.00$ | $0.010 \pm 0.00$ |
| Fashion MNIST | $0.137 \pm 0.01$ | $0.127 \pm 0.01$ | $0.126 \pm 0.00$ | $0.111 \pm 0.00$ | $0.121 \pm 0.00$ | $0.096 \pm 0.00$ |
| CIFAR-10 | $0.469 \pm 0.01$ | $0.450 \pm 0.01$ | $0.447 \pm 0.01$ | $0.381 \pm 0.01$ | $0.373 \pm 0.00$ | $0.335 \pm 0.00$ |
| CIFAR-100 | $0.795 \pm 0.01$ | $0.786 \pm 0.01$ | $0.782 \pm 0.01$ | $0.768 \pm 0.01$ | $0.723 \pm 0.00$ | $0.695 \pm 0.00$ |
| | | | ECE % | | | |
| MNIST (MLP) | $1.324 \pm 0.16$ | $0.528 \pm 0.12$ | $0.415 \pm 0.10$ | $2.569 \pm 0.17$ | $0.536 \pm 0.08$ | $0.839 \pm 0.08$ |
| MNIST (CNN) | $0.517 \pm 0.07$ | $0.366 \pm 0.08$ | $0.259 \pm 0.06$ | $0.832 \pm 0.06$ | $0.282 \pm 0.04$ | $0.287 \pm 0.05$ |
| Fashion MNIST | $5.269 \pm 0.22$ | $1.784 \pm 0.54$ | $1.098 \pm 0.18$ | $1.466 \pm 0.30$ | $3.988 \pm 0.11$ | $0.942 \pm 0.13$ |
| CIFAR-10 | $11.718 \pm 0.72$ | $4.599 \pm 0.82$ | $1.318 \pm 0.26$ | $7.109 \pm 0.62$ | $8.655 \pm 0.29$ | $8.867 \pm 0.23$ |
| CIFAR-100 | $9.780 \pm 0.69$ | $5.535 \pm 0.50$ | $2.012 \pm 0.31$ | $12.608 \pm 0.59$ | $7.180 \pm 0.48$ | $11.954 \pm 0.29$ |
| | | | Classification Error % | | | |
| MNIST (MLP) | $2.264 \pm 0.22$ | $2.286 \pm 0.24$ | $2.264 \pm 0.22$ | $2.452 \pm 0.14$ | $2.082 \pm 0.10$ | $1.285 \pm 0.05$ |
| MNIST (CNN) | $0.990 \pm 0.13$ | $0.990 \pm 0.12$ | $0.990 \pm 0.13$ | $0.842 \pm 0.06$ | $0.868 \pm 0.06$ | $0.659 \pm 0.03$ |
| Fashion MNIST | $8.420 \pm 0.32$ | $8.522 \pm 0.34$ | $8.420 \pm 0.32$ | $7.692 \pm 0.34$ | $7.734 \pm 0.11$ | $6.508 \pm 0.10$ |
| CIFAR-10 | $33.023 \pm 0.68$ | $32.949 \pm 0.74$ | $33.023 \pm 0.68$ | $27.207 \pm 0.66$ | $26.004 \pm 0.36$ | $22.880 \pm 0.21$ |
| CIFAR-100 | $64.843 \pm 0.69$ | $64.789 \pm 0.69$ | $64.843 \pm 0.69$ | $60.772 \pm 0.58$ | $58.092 \pm 0.42$ | $53.917 \pm 0.30$ |

## 3.2 MNIST and CIFAR experiments

The MNIST handwritten digits [27] and fashion MNIST [47] datasets consist of 60,000 training images and 10,000 test images. The CIFAR-10 and CIFAR-100 datasets [24] consists of 50,000 training images and 10,000 test images. We created validation sets by setting aside 10,000 and 5,000 training images from MNIST (handwritten and fashion) and CIFAR, respectively. For the handwritten MNIST dataset, the predictive uncertainty was evaluated for two different neural networks: a Multi-layer Perception (MLP) and a Convolutional Neural Network (CNN) similar to LeNet [29] but with smaller kernel sizes. The MLP is similar to the one used in [26] and has 3 hidden layers with 200 neurons each, ReLu non-linearities, and BN after each layer. For MCD experiments, dropout layers were added after each hidden layer with 0.5 dropout rate as was suggested in [8]. The CNN for MNIST (handwritten and fashion) experiments has two convolutional layers with 32 and 64 kernels of sizes $3 \times 3$ with stride size of 1 followed by two fully connected layers (with 128 and 64 neurons each) with BN after both types of layers. Here, again for MCD experiments, dropout was added after all layers with 0.5 dropout rate, except the first and last layers. For the CIFAR-10 and CIFAR-100 dataset, the CNN architecture has 2 convolutional layers with 16 kernels of size $3 \times 3$ followed by a max-pooling of $2 \times 2$; another 2 convolutional layers with 32 kernels of size $3 \times 3$ followed by a max-pooling of size $2 \times 2$ and a dense layers of size 128, and finally, a dense layer of 10 for CIFAR-10 and 100 for CIFAR-100. BN was applied to all convolutional layers. For MCD experiments, dropout was added similar to CNN for MNIST experiments. Each network was trained and evaluated 25 times with different initializations of parameters (weights and biases) and random shuffling of the training data. For optimization, stochastic gradient descent with the Adam update rule [22] was used. Each baseline was trained for 15 epochs. The training was carried out for another 25 rounds with dropout for MCD experiments. Models trained and evaluated with active dropout layers were used for MCD evaluation only, and baselines without dropout were used for the rest of the experiments. The Deep Ensembles method was tested by averaging the output of the 10 baseline models. MCD was tested on 25 models and the performance was averaged over all 25 models. Temperature scaling and PEP were tested on the 25 trained baseline models without dropout and the results were averaged.

Table 2 compares the calibration quality and test errors of baselines, PEP, temperature scaling [12], MCD [8], Stochastic Weight Averaging (SWA) [19], and Deep Ensembles [26]. The averages and standard deviation values for NLL, Brier score, and ECE% are provided. For all cases, it can be seen that PEP achieves better calibration in terms of lower NLL compared to the baseline. Deep Ensembles

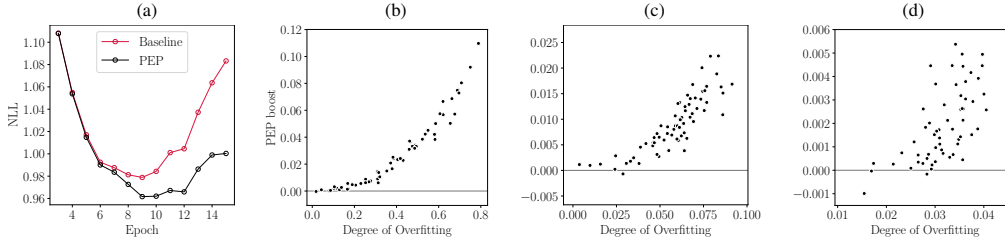

Figure 3: The relationship between overfitting and PEP effect. (a) shows the average of NLLs on test set for CIFAR-10 baselines (red line) and PEP $\mathbb{L}$ (black line). The baseline curve shows overfitting as a result of overtraining. The degree of overfitting was calculated by subtracting the training NLL (loss) from the test NLL (loss). PEP reduces the effect of overfitting and improves log-likelihood. The PEP effect is more substantial as the overfitting grows. (b), (c), and (d) show scatter plots of overfitting vs PEP effect for CIFAR-10, MNIST(MLP), and MNIST(CNN), respectively.

achieves the best NLL and classification errors in all the experiments. Compared to the baseline, temperature scaling and MCD improve calibration in terms of NLL for all three experiments.

**Non-Gaussian Distributions** We performed limited experiments to test the effect of of using non-Gaussian distributions. We tried perturbing by a uniform distribution with MNIST (MLP) and observed similar performance to a normal distribution. Further tests with additional benchmarks and architectures are needed for conclusive findings.

**Effect of Overfitting on PEP effect** We ran experiments to quantify the effect of overfitting on PEP effect, and optimized $\sigma$ values. For the MNIST and CIFAR-10 experiments, model checkpoints were saved at the end of each epoch. Different levels of overfitting as a result of over-training were observed for the three experiments. $\sigma^*$ was calculated for each epoch and PEP was performed and the PEP effect was measured. Figure 3 (a), shows the effect of calibration and reducing NLL for CIFAR-10 models. Figures 3 (b-d) shows that PEP effect increases with overfitting. Furthermore, we observed that the $\sigma^*$ values also increase with overfitting, meaning that larger perturbations are required for more overfitting.

**Out-of-distribution detection** We performed experiments similar to Maddox et al. [32] for out-of-distribution detection. We trained a WideResNet-28x10 on data from five classes of the CIFAR-10 dataset and then evaluated on the whole test set. We measured the symmetrized Kullback–Leibler divergence (KLD) between the in-distribution and out-of-distributions samples. The results show that using PEP, KLD increased from 0.47 (baseline) to 0.72. In the same experiment temperature scaling increased KLD to 0.71.

# 4 Conclusion

We proposed PEP for improving calibration and performance in deep learning. PEP is computationally inexpensive and can be applied to any pre-trained network. On classification problems, we show that PEP effectively improves probabilistic predictions in terms of log-likelihood, Brier score, and expected calibration error. It also nearly always provides small improvements in accuracy for pre-trained ImageNet networks. We observe that the optimal size of perturbation and the log-likelihood increase from the ensemble correlates with the amount of overfitting. Finally, PEP can be used as a tool to investigate the curvature properties of the likelihood landscape.

# 5 Acknowledgements

Research reported in this publication was supported by NIH Grant No. P41EB015898, Natural Sciences and Engineering Research Council (NSERC) of Canada and the Canadian Institutes of Health Research (CIHR).

## 6 Broader Impact

Training large networks can be highly compute intensive, so improved performance and calibration by ensembling approaches that use additional training, e.g., deep ensembling, can potentially cause undesirable contributions to the carbon footprint. In this setting, PEP can be seen as a way to reduce training costs, though prediction time costs are increased, which might matter if the resulting network is very heavily used. Because it is easy to apply, and no additional training (or access to the training data) is needed, PEP provides a safe way to tune or improve a network that was trained on sensitive data, e.g., protected health information. Similarly, PEP may be useful in competitions to gain a mild advantage in performance.

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
