[Supplementary Material · NeurIPS_2020_PEP_Appendix.pdf]

# A  Further Analysis

## A.1  Approximation of Expectation of Non-Linear Function

Suppose $x \sim N(\mu, \Sigma)$. We seek a local approximation to $\mathbb{E}_x\left[f(x)\right]$. Using a second order Taylor expansion about $\mu$,

$$\mathbb{E}_x\left[f(x)\right] \approx \mathbb{E}_x\left[f(\mu) + (x-\mu)^T \nabla f(\mu) + \frac{1}{2}(x-\mu)^T H f(\mu)(x-\mu)\right] \tag{1}$$

where $Hf(x)$ is the Hessian of $f(x)$. Then, as the gradient term vanishes,

$$\mathbb{E}_x\left[f(x)\right] \approx f(\mu) + \frac{1}{2}\mathbb{E}_x\left[(x-\mu)^T H f(\mu)(x-\mu)\right] \tag{2}$$

$$\mathbb{E}_x\left[f(x)\right] \approx f(\mu) + \frac{1}{2}\mathbb{E}_x\left[x^T H f(\mu)x - 2x^T H f(\mu)\mu + \mu^T H f(\mu)\mu\right] \tag{3}$$

or

$$\mathbb{E}_x\left[f(x)\right] \approx f(\mu) + \frac{1}{2}\left[\mathbb{E}_x\left[x^T H f(\mu)x\right] - \mu^T H f(\mu)\mu\right] . \tag{4}$$

Now using $\mathbb{E}_x\left[x^T \Lambda x\right] = T_R(\Lambda\Sigma) + \mu^T \Lambda \mu$,($T_R$ is the trace, see [1]),

$$\mathbb{E}_x\left[f(x)\right] \approx f(\mu) + \frac{1}{2}T_R(H f(\mu)\Sigma) . \tag{5}$$

 **A.1.1 Third and Fourth Moments**

In the following we analyze expectations of the third and fourth Taylor expansion terms, showing that the third term vanishes, and that the fourth is proportional to $\sigma^4 \sum_{1 \leq i,j \leq N} \partial_i^2 \partial_j^2 f(\mu)$. We will refer to the terms schematically as $\mathrm{Taylor}_\mu^3 f(x)$ and $\mathrm{Taylor}_\mu^4 f(x)$. We use $x \sim N(\mu, \Sigma = \sigma^2 I)$ as in Sec. 2.3 of the paper. This implies that for any given $i$: $x_i \sim N(\mu_i, \sigma^2)$; $\mathbb{E}_{x_i}\left[(x_i - \mu)^3\right] = 0$; and $\mathbb{E}_{x_i}\left[(x_i - \mu)^4\right] = 3\sigma^4$. For convenience, in the following derivations nonzero constants of each term of the Taylor series have been omitted, and we denote $\frac{\partial^n}{\partial^n x_i}$ as $\partial_i^n$, ommiting the superindex for $n = 1$.

**Third Moment**

$$\mathbb{E}_x\left[\mathrm{Taylor}_\mu^3 f(x)\right] \propto \sum_{1 \leq i \leq j \leq k \leq N} \mathbb{E}_x\left[\partial_i \partial_j \partial_k f(x)(x_i - \mu_i)(x_j - \mu_j)(x_k - \mu_k)\right] \tag{6}$$

$$\text{linearity of } \mathbb{E} = \sum_{1 \leq i \leq N} \mathbb{E}_x\left[\partial_i^3 f(\mu)(x_i - \mu_i)^3\right] \tag{7}$$

$$+ \sum_{1 \leq i \neq j \leq N} \mathbb{E}_x\left[\partial_i^2 \partial_j f(\mu)(x_i - \mu_i)^2(x_j - \mu_j)\right] \tag{8}$$

$$+ \sum_{1 \leq i \neq j \neq k \leq N} \mathbb{E}_x\left[\partial_i \partial_j \partial_k f(\mu)(x_i - \mu_i)(x_j - \mu_j)(x_k - \mu_k)\right] \tag{9}$$

$$\text{linearity of } \mathbb{E} = \sum_{1 \leq i \leq N} \partial_i^3 f(\mu) \mathbb{E}_x\left[(x_i - \mu_i)^3\right] \tag{10}$$

$$+ \sum_{1 \leq i \neq j \leq N} \partial_i^2 \partial_j f(\mu) \mathbb{E}_x\left[(x_i - \mu_i)^2(x_j - \mu_j)\right] \tag{11}$$

$$+ \sum_{1 \leq i \neq j \neq k \leq N} \partial_i \partial_j \partial_k f(\mu) \mathbb{E}_x\left[(x_i - \mu_i)(x_j - \mu_j)(x_k - \mu_k)\right] \tag{12}$$

$$\text{independence } (\Sigma \text{ is } \sigma^2 I) = \sum_{1 \leq i \leq N} \partial_i^3 f(\mu) \mathbb{E}_{x_i}\left[(x_i - \mu_i)^3\right] \tag{13}$$

$$+ \sum_{1 \leq i \neq j \leq N} \partial_i^2 \partial_j f(\mu) \mathbb{E}_{x_i}\left[(x_i - \mu_i)^2\right] \mathbb{E}_{x_j}\left[x_j - \mu_j\right] \tag{14}$$

$$+ \sum_{1 \leq i \neq j \neq k \leq N} \partial_i \partial_j \partial_k f(\mu)$$
$$\cdot \mathbb{E}_{x_i}\left[x_i - \mu_i\right] \mathbb{E}_{x_j}\left[x_j - \mu_j\right] \mathbb{E}_{x_k}\left[x_k - \mu_k\right] \tag{15}$$

$$\text{Due to } x \sim N(\mu, \sigma^2 I) = \sum_{1 \leq i \leq N} \partial_i^3 f(\mu) \cdot 0 \quad \text{(no skew)} \tag{16}$$

$$+ \sum_{1 \leq i \neq j \leq N} \partial_i^2 \partial_j f(\mu) \mathbb{E}_{x_i}\left[(x_i - \mu_i)^2\right] 0 \quad (\mu \text{ mean}) \tag{17}$$

$$+ \sum_{1 \leq i \neq j \neq k \leq N} \partial_i \partial_j \partial_k f(\mu) \cdot 0 \cdot 0 \cdot 0 \quad (\mu \text{ mean}) \tag{18}$$

$$= 0 \tag{19}$$

18 **Fourth Moment** We now derive the fourth moment without most of the tedious algebra we used to
19 derive the third, but following the same ideas.

$$\mathbb{E}_x \left[ \text{Taylor}^4_\mu f(x) \right] \propto$$

$$\sum_{1 \leq i \leq j \leq k \leq l \leq N} \mathbb{E}_x \left[ \partial_i \partial_j \partial_k \partial_l f(x)(x_i - \mu_i)(x_j - \mu_j)(x_k - \mu_k)(x_l - \mu_l) \right] \quad (20)$$

$$= \sum_{1 \leq i \leq N} \partial_i^4 f(\mu) \mathbb{E}_x \left[ (x_i - \mu_i)^4 \right] \quad (21)$$

$$+ \sum_{1 \leq i \neq j \leq N} \partial_i^2 \partial_j^2 f(\mu) \mathbb{E}_{x_i} \left[ (x_i - \mu_i)^2 \right] \mathbb{E}_{x_j} \left[ (x_j - \mu_j)^2 \right] \quad (22)$$

$$+ \sum_{1 \leq i \neq j \leq N} \partial_i^3 \partial_j f(\mu) \mathbb{E}_{x_i} \left[ (x_i - \mu_i)^3 \right] \mathbb{E}_{x_j} \left[ (x_j - \mu_j) \right] \quad (23)$$

$$+ \sum_{1 \leq i \neq j \neq k \leq N} \partial_i^2 \partial_j \partial_k f(\mu)$$
$$\cdot \mathbb{E}_{x_i} \left[ (x_i - \mu_i)^2 \right] \mathbb{E}_{x_j} \left[ x_j - \mu_j \right] \mathbb{E}_{x_k} \left[ x_k - \mu_k \right] \quad (24)$$

$$+ \sum_{1 \leq i \neq j \neq k \neq l \leq N} \partial_i \partial_j \partial_k \partial_l f(\mu)$$
$$\cdot \mathbb{E}_{x_i} \left[ x_i - \mu_i \right] \mathbb{E}_{x_j} \left[ x_j - \mu_j \right] \mathbb{E}_{x_k} \left[ x_k - \mu_k \right] \mathbb{E}_{x_l} \left[ x_l - \mu_l \right] \quad (25)$$

$$= \sum_{1 \leq i \leq N} \partial_i^4 f(\mu) \mathbb{E}_x \left[ (x_i - \mu_i)^4 \right] \quad (26)$$

$$+ \sum_{1 \leq i \neq j \leq N} \partial_i^2 \partial_j^2 f(\mu) \mathbb{E}_{x_i} \left[ (x_i - \mu_i)^2 \right] \mathbb{E}_{x_j} \left[ (x_j - \mu_j)^2 \right] \quad (27)$$

$$= \sigma^4 \left( 3 \sum_{1 \leq i \leq N} \partial_i^4 f(\mu) + \sum_{1 \leq i \neq j \leq N} \partial_i^2 \partial_j^2 f(\mu) \right) \quad (28)$$

$$\propto \sigma^4 \sum_{1 \leq i,j \leq N} \partial_i^2 \partial_j^2 f(\mu) \quad (29)$$

## 20 A.2 Laplacian of Log Likelihood

21 We derive here the Laplacian of the log likelihood of the base network.

$$\triangle \mathcal{L}(\theta) = \sum_k \frac{\partial^2}{\partial \theta_k^2} \sum_i \ln L_i(\theta) = \sum_k \sum_i \frac{\partial^2}{\partial \theta_k^2} \ln L_i(\theta) . \quad (30)$$

22 Differentiating,

$$\triangle \mathcal{L}(\theta) = \sum_k \sum_i \frac{\partial}{\partial \theta_k} \frac{\frac{\partial}{\partial \theta_k} L_i(\theta)}{L_i(\theta)} , \quad (31)$$

23 then

$$\triangle \mathcal{L}(\theta) = \sum_k \sum_i \left[ \frac{\frac{\partial^2}{\partial \theta_k^2} L_i(\theta)}{L_i(\theta)} - \frac{(\frac{\partial}{\partial \theta_k} L_i(\theta))^2}{L_i^2(\theta)} \right] , \quad (32)$$

24 or,

$$\triangle \mathcal{L}(\theta) = \sum_i \left[ \frac{\triangle L_i(\theta)}{L_i(\theta)} - \frac{\nabla L_i(\theta)^2}{L_i^2(\theta)} \right] , \quad (33)$$

25 where $\nabla L_i(\theta)^2 = \nabla L_i(\theta) \cdot \nabla L_i(\theta)$. Then

$$\triangle \mathcal{L}(\theta) = \sum_i \left[ \frac{\triangle L_i(\theta)}{L_i(\theta)} - (\nabla \ln L_i(\theta))^2 \right] . \quad (34)$$

## B Evaluation metrics

For a K-class classification problem, with N samples, NLL and Brier score are calculated as $-N^{-1}\sum_{n=1}^{N}\sum_{k=1}^{K} y_{i,k} \cdot \ln\left(p_{i,k}\right)$ and $-K^{-1}N^{-1}\sum_{n=1}^{N}\sum_{k=1}^{K}(y_{i,k} - p_{i,k})^2$, respectively. Where $y_{i,k}$ is the true one-hot encoded label which is 1 if sample $i$ has label $k \in K$, and otherwise is 0. $p_{i,k}$ is the predicted class probability of sample $i$ belonging to class $k \in K$. Reliability diagrams plot expected accuracy as a function of class probability (confidence). Expected Calibration Error (ECE) is used to summarize the results of reliability diagrams. Details of evaluation metrics are given in the Supplementary Material. For expected accuracy measurement, the samples are binned into N groups and the accuracy and confidence for each group are computed. Assuming $D_m$ to be indices of samples whose confidence predictions are in the range of $\left(\frac{m-1}{M}, \frac{m}{M}\right]$, the expected accuracy of the $D_m$ is $Acc(D_m) = |D_m|^{-1}\sum_{i \in D_m} y_{i,k}$. The average confidence on bin $D_m$ is calculated as $\overline{P}(D_m) = |D_m|^{-1}\sum_{i \in D_m} p_{i,k}$. ECE is calculated by summing up the weighted average of the differences between accuracy and the average confidence over the bins: $\text{ECE} = \sum_{m=1}^{M} N^{-1}|D_m|\big|ACC(D_m) - \overline{P}(D_m)\big|$.

## C Additional Results on ImageNet

Figure 1 shows reliability diagrams together with ECE values for baseline, temperature scaling and parameter ensembling with perturbation (PEP) for the pre-trained ImageNet networks.

## References

[1] Arakaparampil M Mathai and Serge B Provost. *Quadratic forms in random variables: theory and applications*. Dekker, 1992.

Figure 1: Reliability diagrams and ECE values before and after calibration with Temperature Scaling and PEP, for experiments described in Section 3.1 of the manuscript. From top to bottom: DenseNet121, InceptionV3, ResNet50, VGG16, and VGG19.