[Reviews · NeurIPS 2020]

Review 1

Summary and Contributions: The paper proposed a simple ensembling approach that uses random perturbations of the optimal parameters, where the weights of each layer are perturbed to form a candidate model in the ensemble. The proposed method does not require additional training steps. In fact, the proposed ensemble technique tries to maximize the log-likelihood of the average of predictions from each of the perturbed sets of parameters (or model). The proposed method is evaluated on three computer vision datasets: MNIST, CIFAR-10 and ImageNet. The empirical results indicate improved calibration performance of the predictions.

Strengths: 1. Averaging the predictions from multiple perturbations of model params is simple and yet elegant in terms of ease of use. 2. The method is potentially useful for those that are frugal in training large DNNs from scratch. This for sure has some practical impact in a number of real-world applications, for example, in the case of medical data where security and privacy are important thereby limited to using a few compute resources. In such scenarios, the proposed method can be put to great use. 3. Overall, the parameter perturbation is inspired from a number of existing approaches in climate modeling and/or related fields. For example, Murphy, J., Clark, R., Collins, M., Jackson, C., Rodwell, M., Rougier, J.C., Sanderson, B., Sexton, D. and Yokohata, T., 2011, June. Perturbed parameter ensembles as a tool for sampling model uncertainties and making climate projections. In Proceedings of ECMWF Workshop on Model Uncertainty (pp. 183-208). Bellprat, O., Kotlarski, S., Lüthi, D. and Schär, C., 2012. Exploring perturbed physics ensembles in a regional climate model. Journal of Climate, 25(13), pp.4582-4599.

Weaknesses: 1. Perturbing the weights in a way penalizes (some sort of friendly noise to the parameters) thereby the regularization effect and curbing the over-fiting, great that it improves confidence as well. However, one has to be careful with this additive noise in order to keep it as friendly noise rather than becoming adversarial noise. Looks like the golden search did take care of that with a good enough sigma, moreover, the classification errors also reduced on ImageNet. If the classification improvements are reported on all the benchmarks, that could provide enough empirical evidence. 2. Overall, the empirical evidence and the comparisons look interesting, however, having results on more benchmarks such as CIFAR-100, STL-10, Fashion MNIST (results on all or some of these benchmarks) would further improve the confidence. 3. Referred to the wrong equation “From Appendix (Eq 11)” 4. “Details of evaluation metrics are given in the Supplementary Material.”, please direct the reader to a particular section in Appendix, like, (refer to Appendix B) 5. The MLP and CNN are a bit old models. It is understood that for the sake of proof of concept, the selection of those architectures is fine; however, advanced architectures could be more useful to evaluate the effectiveness of the proposed approach. 6. It is understandable that the comparative methods are ensembles and/or are made to act as ensembles (for example, TS with 10 models). However, Ref. [39] in the bibliography reported better results (on Imagenet and CIFAR10 and other benchmarks not part of this paper) with mixup in terms of ECE compared to TS, label smoothing, and entropy regularized loss. 7. The Gaussian prior for perturbations seems a natural choice given that the distribution of DNN parameters might belong to the same distribution. However, what happens if it uniform distribution prior or some other distribution? My first guess is not going to work, however want to confirm once. The idea is simple and interesting, having a few more experiments can highlight the strength of the proposed approach. Addressing some or all of the above concerns can help improve the scores.

Correctness: Yes, to the best of my understanding and knowledge, the empirical results and the claims are correct.

Clarity: Yes, overall, the paper is well written and easy to follow.

Relation to Prior Work: The proposed method makes a clear distinction with the state-of-the-art methods in literature. The proposed method, PEP, takes advantage of training once as opposed to Deep Ensembles and no need of dropouts as opposed to monte carlo dropouts.

Reproducibility: Yes

Additional Feedback: Another interesting empirical evidence is the performance of PEP on out-of-distribution images. How does it perform, does it hold the predictions with good uncertainty? May be results on a couple of benchmarks would highlight the strengths of the proposed approach. Nonetheless, this should not influence in giving low scores, certainly will help to increase the review score. Post rebuttal comments All the concerns are addressed and are satisfactory. Therefore increasing the score to clear accept.


Review 2

Summary and Contributions: The authors introduce a method for creating a cheap ensemble from an existing pre-trained network.The authors sample new network parameters from an isotopic Gaussian centered around the pre-trained parameters, where the Gaussian’s variance is found through cross-validation. The authors show that such ensembles can result in better likelihood and calibration, especially for models prone to overfitting.

Strengths: This paper is extremely well-written, and the procedure is simple and well analyzed. The authors include theoretical and empirical analysis, and their experiments explore and demonstrate when their method is most effective. The procedure is very simple and, because it can be applied to pre-trained networks without modification, could easily be used in practice. ============= Thank you for a very thorough rebuttal. As a quick follow up, here is another (recent) paper that is performing a similar Laplace approximation: https://arxiv.org/abs/2002.10118 I recognize that this paper is ICML 2020 - and therefore there was no reason to expect that it should've been referenced in the original submission. (It is not influencing my review at all.) However, since it is relevant it would be worth mentioning in your revisions.

Weaknesses: While overall I think the paper is well written and executed, there are many missing references to very relevant and related pieces of work. The proposed method is very similar to a Laplace approximation (from the approximate Bayesian inference community), which has been explored in the context of neural networks by Ritter et al. (ICLR, 2018). Those authors propose an extremely similar method (though in the context of Bayesian machine learning): approximating the distribution of neural network parameters with an isotopic Gaussian, and evaluating this ensemble through sampling. The procedure proposed by this paper may be more practical than Ritter’s procedure (as it uses a simple cross validation to determine the noise parameter), and it is targeted in a slightly different context (neural network ensembles verses approximate Bayesian inference). Yet at the same time, in the context of this other paper the novelty of PEP may be limited. Moreover, there is existing work that constructs cheap “perturbed” ensembles using the inherent randomness of SGD (e.g. snapshot ensembles, stochastic weight averaging - Izmailov et al., UAI 2018 and Maddox et al., NeurIPS 2019). I would encourage the authors to the Stochastic Weight Averaging - Gaussian method of Maddox et al. in the CIFAR experiments, as it is a very relevant baseline.

Correctness: The empirical analysis is very thourough and careful. The authors clearly outline their methodology and, due to the simplicity of their method, clearly demonstrate its effectiveness. While the theoretical analysis is all correct, much of it is also well established (second order Taylor expansions as Laplace approximations, the odd moments of zero-mean Gaussians are zero, etc).

Clarity: The paper is extremely well written and clear, though I had some confusion with the paragraph starting at line 161. I’m not sure what you mean by a “different empirical FI?” Also, given that there are many equations in succession on page 5, it is very confusing what “first term” and “second term” refer to (are you talking about the terms from equation 16, or the most recent equation?).

Relation to Prior Work: As stated above, this paper is missing some key references to recent related work. It is very similar to what was proposed by Ritter et al. (2018), and the SWA methods of Izmailov et al. + Maddox et al. should be included as baselines. In the discussion of PEP effect vs. overfitting - there is existing work that shows how network generalization behaves when parameters are perturbed (see, for example, Goodfellow et al. (ICLR, 2015 “Qualitatively Characterizing Neural Network Optimization Problems”). This may be a good paper to include in the related work.

Reproducibility: Yes

Additional Feedback: No broader impact section.


Review 3

Summary and Contributions: The paper presents an approach (called PEP) for improving calibration and performance in deep learning, which is inexpensive and can be applied to any pretrained network. On classification problems, PEP effectively improves probabilistic predictions in terms of log-likelihood, Brier score, and expected calibration error. It also provides small improvements in accuracy for pretrained ImageNet networks.

Strengths: This work is novel and potentially useful. It is certainly also relevant to NeurIPS. The claims are theoretically and experimentally grounded.

Weaknesses: The performance of the approach is still much worse than for two competing approaches.

Correctness: The contents seems to be correct.

Clarity: The paper is very well written.

Relation to Prior Work: The paper contains a satisfactory discussion of related work.

Reproducibility: Yes

Additional Feedback: I have read the rebuttal.


Review 4

Summary and Contributions: The authors propose PEP to increase predictive performance and calibration in Deep Networks. This approach uses an ensemble of parameter values as perturbations of the optimal parameters, given by a Gaussian with a variance parameter. The set of best parameters is chosen on the training set, and the variance is chosen to maximize the likelihood on the validation set.

Strengths: The authors state the strengths of their methods as opposed to other methods used to improve calibration and uncertainty estimation: unlike other methods, theirs can be applied to any pre-trained network without restrictions, and needs only one training run.

Weaknesses: It seems that since the paper’s goal is to improve calibration and performance, the evaluation measures that should matter are ECE and the classification error. However, PEP does not seem to necessarily improve these measures.

Correctness: The claims and method seem to be correct.

Clarity: The paper is explained well, but the notation, while clear, is not always defined. For example, please state clearly what i, j, and m are.

Relation to Prior Work: Yes, difference from previous works is clearly discussed.

Reproducibility: Yes

Additional Feedback: The paper doesn’t mention the reason this specific approach was chosen; i.e., please given an intuitive explanation as to why parameter perturbations set from training by a Gaussian would help with calibration. ============= Thank you for your thorough response, I raised my score to 5 (still concerned about the evaluation).

[Author Response · NeurIPS 2020]

We thank the reviewers for their thoughtful and constructive reviews. We appreciate that they found the paper to be
'theoretically and experimentally grounded,' and 'extremely well-written,' that it 'could easily be used in practice,' and
has 'practical impact in a number of real-world applications,' specifically 'where security and privacy are important.'
Below we respond to the major comments; we will fix the minor ones in the final version.
**Reviewer #1** ● *results on more benchmarks [...] would further improve the confidence.* Agreed. As suggested, we
ran experiments on the Fashion MNIST benchmark; The results (Table I) are substantially in line with those in the
paper. Experiments for CIFAR-100 are in progress and we will add those and the results of Table I to the final paper.
● *The MLP and CNN are a bit old models [...]* We used MLP and CNNs since they were used in studies that we
compared to, e.g. Deep Ensembles (DE) (arXiv:1612.01474). Furthermore, in the submitted paper, we showed the
effectiveness of our proposed methods on larger and deeper pretrained networks. In the new sets of experiments, we
used Wide-ResNet-28×10 (arXiv:1605.07146v4 2016) for CIFAR-10 out-of-distribution (OOD) detection experiments.
● *Ref. [39] reported improved results [...] using 'mixup'* Agreed. Due to the limited time of the response period, that
experiment cannot be completed now. We will add the 'mixeup training' results into the revised paper. ● *uniform*
*distribution prior or some other distribution [instead of normal for perturbation] ....?* We tested a uniform distribution
with MNIST (MLP) and observed similar performance to a normal distribution on this small problem. We will run
experiments on other applications, including larger ImageNet networks, and add the results to the final version of the
paper. ● *the performance of PEP on out-of-distribution images.* We performed experiments similar to arXiv:1902.02476
for OOD detection. We trained a WideResNet-28x10 on data from five classes of the CIFAR-10 dataset and then
evaluated on the whole test set. We measured the symmetrized KL divergence (KLD) between the in-distribution and
out-of-distributions samples. The results show that KLD increased from 0.47 (baseline) to 0.72 by using PEP. Temp.
scaling also increased KLD to 0.71. We will add these results to the paper.

**Table 1:** Aditional experiments on fashion MNIST (For all metrics smaller is better).

| Metric | Baseline | PEP | Temp. Scaling | MCD | Deep Ensembles |
|---|---|---|---|---|---|
| NLL | $0.360 \pm 0.01$ | $0.275 \pm 0.01$ | $0.271 \pm 0.01$ | $0.218 \pm 0.01$ | $0.198 \pm 0.00$ |
| Brier | $0.137 \pm 0.01$ | $0.127 \pm 0.01$ | $0.126 \pm 0.00$ | $0.111 \pm 0.00$ | $0.096 \pm 0.00$ |
| ECE % | $5.269 \pm 0.22$ | $1.784 \pm 0.54$ | $1.098 \pm 0.18$ | $1.466 \pm 0.30$ | $0.942 \pm 0.13$ |
| Classification Error | $8.420 \pm 0.32$ | $8.522 \pm 0.34$ | $8.420 \pm 0.32$ | $7.692 \pm 0.34$ | $6.508 \pm 0.10$ |

**Reviewer #2** ● *there are many missing references to very relevant and related pieces of work.* We thank R2 for
pointing out the related work of (Ritter et al. 2010), (Izmailov et al. 2018) and (Maddox et al. 2019), especially about
Laplace approximations. From that point of view, PEP is perhaps the simplest possible Laplace approximation - an
isotropic Gaussian with one variance parameter, though we set the parameter with simple ML/cross-validation rather
than calculating curvature. The trade-off is that while performance is expected to be better with richer covariance
models, there is some overhead in calculating them, and they are not practical for use with pre-trained models. We will
revise the paper accordingly. ● *In the discussion of PEP effect vs. overfitting [... (Goodfellow ICLR 2015) ...] may be a*
*good paper to include in the related work.* Agreed. We will include it in the final version. ● *SWA methods of Izmailov*
*et al. + [SWAG method] Maddox et al. should be included as baselines.* We agree that addition of SWA/G results will
strengthen the conclusions. We are addressing the implementation logistics between us and SWA/G and (SWAG, more
recent, is in PyTorch, SWA TensorFlow implementation has bugs, we are in TensorFlow) which need additional time.
We will add results of the experiments in the revision. ● *No broader impact section* We will add a 'broader impact'
section that will discuss the importance of reducing carbon footprint via reduced compute resources, and the importance
of improved calibration, security and privacy in medical applications. ● *While the theoretical analysis is all correct,*
*much of it is also well established (... Taylor expansions as Laplace approximations ... odd moments of ... Gaussians*
*are zero, etc).* We agree that we are using well-established methods; we think of this as an advantage. Our work shows
how a simple formalism yields improvement in the calibration of pre-trained networks. It also enables us to provide an
in-depth analysis of why our method can improve NLL, and under what conditions. ● *[what are] "different empirical*
*FI?" ... "first term" ... "second term"* FI is Fisher Information, 'First term' and 'Second term' refer to the preceding
equation. We will clarify in the revision accordingly.
**Reviewer #3** Thank you for appreciating the novelty of our approach. ● *The performance [...] is still much worse than*
*for two competing approaches.* True, but DE has additional training cost, and MCD requires model modification.
**Reviewer #4** ● *the evaluation measures that should matter are ECE and the classification error. However, PEP does*
*not seem to necessarily improve these measures.* PEP is mainly aimed at improving calibration, though it can provide
classification improvement for overfitted models. NLL, Brier score, and ECE are all commonly used metrics to assess
calibration. PEP improved ECE of the baselines in 8 out of 9 experiments. It also consistently improved NLL and Brier
score of the baselines. ● *The paper doesn't mention the reason this specific approach was chosen; i.e., please given an*
*intuitive explanation [...]* There are general arguments about why ensembles can work [ref 5]. Also, Jensen's inequality
suggested to us that simple probabilistic perturbations about $\theta^*$ might be effective, depending on the curvature of the
validation log likelihood function at $\theta^*$ (from training), (which might depend on overfitting). ● *notation, while clear, is*
*not always defined. please state clearly what i, j, and m are.* They are index of data item, index of Gaussian sample,
number of Gaussian samples, respectively. We will revise the paper accordingly.

[Meta-Review · NeurIPS 2020]

Three knowledgeable referees support acceptance for the contributions, given the simplicity, effectiveness and theoretical soundness of the proposed approach. However, please consider revising the paper to address R1 and R2's remark on - highlighting the connection with prior work on Baysian inference such as Ritter et al. 2018. - comparison with stochastic based ensembling approach such as Stochastic Weight Averaging (SWA). - Evaluation results on out-of-distribution detection tasks.